# Factors Related to Hospitalisation-Associated Disability in Patients after Surgery for Acute Type A Aortic Dissection: A Retrospective Study

**DOI:** 10.3390/ijerph191912918

**Published:** 2022-10-09

**Authors:** Kotaro Hirakawa, Atsuko Nakayama, Masakazu Saitoh, Kentaro Hori, Tomoki Shimokawa, Tomohiro Iwakura, Go Haraguchi, Mitsuaki Isobe

**Affiliations:** 1Department of Rehabilitation, Sakakibara Heart Institute, Tokyo 183-0003, Japan; 2Department of Cardiology, Sakakibara Heart Institute, Tokyo 183-0003, Japan; 3Department of Physical Therapy, Faculty of Health Science, Juntendo University, Tokyo 113-0033, Japan; 4Department of Cardiovascular Surgery, Sakakibara Heart Institute, Tokyo 183-0003, Japan; 5Division of Intensive Care Unit, Sakakibara Heart Institute, Tokyo 183-0003, Japan; 6Sakakibara Heart Institute, Tokyo 183-0003, Japan

**Keywords:** acute type A aortic dissection, hospitalisation-associated disability, emergency surgery, rehabilitation

## Abstract

The in-hospital mortality rate among patients after surgery for acute type A aortic dissection (ATAAD) has improved chronologically. However, the relationship between the incidence of hospitalisation-associated disability (HAD) and acute cardiac rehabilitation in patients after surgery for ATAAD has not been reported. Therefore, this study evaluated factors related to HAD in patients after surgery for ATAAD. This single-centre retrospective observational study included 483 patients who required emergency surgery for ATAAD. HAD occurred in 104 (21.5%) patients following cardiovascular surgery. Factors associated with HAD were age (odds ratio [OR], 1.05; 95% confidence interval [CI], 1.02–1.09; *p* = 0.001), noninvasive positive pressure ventilation (NPPV; OR, 2.15; 95% CI, 1.10–4.19; *p* = 0.025), postoperative delirium (OR, 2.93; 95% CI, 1.60–5.37; *p* = 0.001), and timing of walking onset (OR, 1.29; 95% CI, 1.07–1.56; *p* = 0.008). Furthermore, a late walking onset was associated with a higher risk of developing HAD and more severe functional decline. Early rehabilitation based on appropriate criteria has possibility of preventing HAD.

## 1. Introduction

Acute type A aortic dissection (ATAAD) requires urgent highly invasive surgery because it leads to rapid post-onset haemodynamic deterioration such as severe circulatory failure due to cardiac tamponade, aortic rupture, and multiple organ failure caused by malperfusion [1]. The in-hospital mortality rate of patients after surgery for ATAAD has improved over time due to rapid treatment decisions influenced by the establishment of diagnostic imaging protocols, development of surgical techniques, and progression of perioperative management [2]. ATAAD involving dissection of the ascending aorta has a very poor prognosis for medical therapy, whereas the one-year survival rate after surgery is reported to be 95% [3].

Hospitalisation-associated disability (HAD), which implies a decline in physical function and activities of daily living (ADL), have been reported to impact the quality of life (QOL) and long-term prognosis of patients after discharge [4,5]. HAD can affect their discharge from the hospital and may require the introduction of new medical resources. Therefore, HAD is an important outcome at discharge in patients who have undergone cardiovascular surgery such as CABG, Valve, and Thoracic aortic surgery. It is associated with preoperative factors such as age, cognitive function, prehospital lifestyle, the environment, and inpatient treatment factors such as bed rest, malnutrition, and polypharmacy [6,7]. After surgery for ATAAD, patients are expected to be at high risk of developing HAD because of hypermetabolism caused by the surgical invasion and systemic inflammation leads to muscle protein degradation and muscle dysfunction [8,9]. In addition, they require postoperative bed rest. However, the incidence of HAD or HAD-related factors in patients after surgery for ATAAD has not been reported. To prevent HAD, cardiac rehabilitation after cardiovascular surgery aims to restore preoperative physical function as early as possible. Early rehabilitation has been shown to be effective in improving physical function and health-related quality of life (QOL) and shortening the duration of mechanical ventilation, the length of stay in the intensive care unit (ICU), and hospital stay [10], and it is expected to be effective in preventing HAD. Currently, there is no evidence on the effectiveness of cardiac rehabilitation after surgery for ATAAD, and the progression of acute rehabilitation and its effects have not been elucidated.

In this study, we evaluated the factors related to HAD in patients after surgery for ATAAD.

## 2. Materials and Methods

### 2.1. Participants

This study was retrospectively conducted at a single cardiovascular centre with the largest number of cardiovascular surgeries in Japan (1550 cardiovascular surgeries in 2020) [11]. This study included 483 patients (median age, 69 years; 52% female) who required emergency surgery for ATAAD at our hospital from April 2014 to December 2020 (Approval ID: 20-010). ATAAD was defined as the Stanford classification type A with dissection including the ascending aorta within 2 weeks of onset. The traumatic type A acute aortic dissection is not included. The exclusion criteria were ADL non-self-reliance, postoperative cerebral infarction or spinal cord infarction, transfer within seven days after surgery, and in-hospital death.

### 2.2. Definition of HAD

The Barthel index (BI) was used to assess ADL levels twice: preoperatively and at discharge. HAD was defined as a decrease in the BI score of at least 5 points at discharge from the score obtained preoperatively [12]. The BI consists of 10 items including (1) feeding, (2) moving from wheelchair to bed and return, (3) personal toilet, (4) getting on and off the toilet, (5) bathing, (6) transferring, (7) ascending and descending stairs, (8) dressing, (9) controlling bowels, and (10) controlling the bladder. The degree of independence was scored as 0, 5, 10, or 15 points, and the total score was calculated for each patient [13]. The preoperative ADL level, obtained either directly from the patient or from relatives, was regarded as a condition that was not exacerbated before admission. In addition, the BI change was categorised as mild with a decrease of 5 points, moderate with a decrease of 10 to 20 points, and severe with a decrease of 25 points or more. Patients with HAD were compared with patients without HAD (non-HAD).

### 2.3. Postoperative Cardiac Rehabilitation

Postoperative cardiac rehabilitation was performed in compliance with the Guidelines for Rehabilitation in Cardiovascular Diseases established in 2012 with the inception and discontinuation criteria [14], and in consultation with attending physicians. The days on which a patient was able to sit, stand, and practice walking independently were referred to as the onset of each of these activities. In addition, the day of walking onset was categorised as early-start walking (within 2 days), usual-start walking (3–4 days), and delayed-start walking (5 days or more). To elucidate the factors that hinder early mobilisation, we investigated the primary reason (sedation, not awakening, pain distress, uncontrolled blood pressure, arrhythmia, respiratory-related, active bleeding, acute kidney injury, acute limb ischemia, and others) for the inability to stand within 48 h of surgery.

### 2.4. Additional Assessments

The following data were collected from medical records: age, sex, body mass index, medical history of comorbidities, preoperative blood biochemistry examination results, intraoperative records, Acute Physiology and Chronic Health Evaluation (APACHE) II score, duration of ventilator intubation, continuous renal replacement therapy (CRRT), noninvasive positive pressure ventilation (NPPV), postoperative complications, length of ICU stay, and length of hospital stay. Acute kidney injury was defined as a creatinine criterion of the Kidney Disease: Improving Global Outcomes (KDIGO) guidelines for acute kidney injury [15]. Postoperative delirium was assessed in the ICU using the confusion assessment method [16].

### 2.5. Statistical Analysis

Continuous variables are expressed as median (interquartile range [IQR]), and categorical variables are expressed as percentages. The Shapiro-Wilk test was used to verify a normal distribution. In group comparisons, the Mann-Whitney U test and Kruskal-Wallis test were performed for continuous variables, and the chi-squared test was used for categorical variables. Multivariate logistic regression analysis was performed, with the presence of HAD as the dependent variable and multiple continuous and categorical variables as independent variables. Age, sex, albumin level, haemoglobin level, duration of ventilator intubation, APACHE II score, CRRT, NPPV, delirium, length of stay in the ICU, and the day of walking onset that were significantly different in the univariate analysis were included as adjusted variables. The level of significance was set at *p* ˂ 0.05, and all statistical analyses were performed using IBM SPSS Statistics version 22 (IBM Corp., Armonk, NY, USA).

## 3. Results

Of the 483 patients included in the analysis, 104 (21.5%) had HAD (Figure 1). The preoperative clinical characteristics, intraoperative findings, and postoperative course of the patients are shown in Table 1. Compared with the non-HAD group, the HAD group had a significantly delayed progression of postoperative cardiac rehabilitation (*p* < 0.001). Of the 483 patients, 131 patients were unable to stand within 48 h, mostly because they were not awakened (39%), or due to respiratory-related reasons (20%) (Table 2). According to walking categories, the delayed-start walking group had a longer operation time, ventilator intubation time, bleeding volume, and length of ICU stay and a higher incidence of postoperative complications than those observed in the early- and usual-start walking groups (Table 3). Although the BI scores were not significantly different between the HAD and non-HAD groups before surgery, they were significantly lower in the HAD group than in the non-HAD group at discharge (*p* < 0.001). The BI scores sub-items tended to decrease with stair climbing, bathing, and walking in more than 50% of HAD patients (Figure 2). Among the patients with HAD, the rate of decrease in BI score by 25 points or above was 22% in the early-start and usual-start walking groups, while it was 61% in the delayed-start walking group. The BI score decreased significantly in the delayed-start walking group (Figure 3).

Univariate logistic regression analysis revealed age, sex, albumin level, haemoglobin level, duration of ventilator intubation, APACHE II score, CRRT, NPPV, delirium, length of stay in the ICU, and timing of walking onset as independent variables for HAD. Of these factors, age (odds ratio [OR], 1.05; 95% confidence interval [CI], 1.02–1.09; *p* < 0.01), NPPV (OR, 2.15; 95% CI, 1.10–4.19; *p* < 0.05), postoperative delirium (OR, 2.93; 95% CI, 1.60–5.37; *p* < 0.01), and timing of walking onset (OR, 1.29; 95% CI, 1.07–1.56; *p* < 0.01) were identified as relevant factors for HAD (Table 4). Moreover, the delayed-start walking group had a higher risk of HAD than the early-start walking group, even after adjusting for confounding variables (OR, 2.76; 95% CI, 1.05–7.21; *p* < 0.05) (Table 5).

## 4. Discussion

We evaluated factors related to HAD in patients who had undergone surgery for ATAAD. The incidence of HAD after surgery for ATAAD was 21.5%, and age, NPPV, postoperative delirium, and timing of walking onset were relevant factors for HAD. Furthermore, a delay in the start of walking was associated with a higher risk of developing HAD and a more severe functional decline.

### 4.1. The Progression of Cardiac Rehabilitation after ATAAD Surgery

The progress of cardiac rehabilitation was carried out by a multi-professional team including a cardiovascular surgeon, an intensivist, and a nurse, led by a physical therapist. ATAAD requires urgent highly invasive surgery because it leads to rapid post-onset haemodynamic deterioration. Postoperatively, patients with ATAAD need strict blood pressure management due to various pathological conditions, such as distal dissection and multiple organ failure due to malperfusion [1]. Postoperative rehabilitation after surgery for ATAAD has rarely been reported and has not yet been established because treatment strategies differ depending on the patient’s medical condition. After surgery for ATAAD, postoperative complications such as cerebral infarction and spinal cord infarction, which inhibit the progress of rehabilitation and cause physical dysfunction, are likely to occur [1]. Nevertheless, in this study, patients with postoperative cerebral infarction or spinal cord infarction were excluded, and HAD was not shown to be caused by cerebral or spinal cord infarction, which has a significant impact on physical function. Early mobilisation is often difficult due to unawakening and respiratory-related problems after surgery. Postoperative unawakening often occurs in ATAAD after surgery due to cerebral perfusion defects at onset and the effects of intraoperative circulatory arrest on cerebral perfusion [17]. This often precludes early mobilisation as a condition of non-consensus. In addition, ATAAD is associated with an increased risk of developing acute respiratory dysfunction, including emergency surgery, a highly invasive systemic inflammatory response due to the use of cardiopulmonary bypass, and complications of organ perfusion disorders [18,19]. In this study, we speculated that these negative influences inhibited early mobilisation in patients with ATAAD after surgery. Patients with delayed walking onset had longer operation times and developed postoperative complications, including multiple organ failure, and some patients with more severe postoperative conditions required systemic management in the ICU. Patients with ATAAD require urgent and highly invasive surgeries. Therefore, it has been suggested that the progress of postoperative rehabilitation is significantly influenced by surgery and the duration of the perioperative period. Furthermore, it is important to begin systemic management from the early postoperative period in critically ill patients to stabilise their condition and ensure early rehabilitation. After surgery for ATAAD, reoperation may be required because of a false lumen or enlargement of the distal aortic diameter [20]. Early rehabilitation can cause enlargement or re-dissection of the distal region. However, in this study, enlargement or re-dissection of the distal region occurred in only 7% of the patients, and none required intervention. Early rehabilitation under strict haemodynamic management has been suggested to be safe.

### 4.2. Factors Associated with HAD

In this study, the incidence of HAD was 21.5%, and age, NPPV, postoperative delirium, and the timing of walking onset were identified as associated factors. There have been many previous reports of age-related decline in ADL [21,22]. HAD develops in 30% of elderly individuals aged 70 and above due to hospitalisation for acute illness, and it has been shown that age-related declines in physical reserve and mental dysfunction, such as cognitive dysfunction and depression, are risk factors for HAD [6,23]. The incidence of HAD in this study was the same as previously reported in patients with postoperative cardiac surgery [24]. In this study, the risk of HAD was significantly higher in patients aged >80 years. Because the general physical ability of the older adult population is increasing in Japan [25], it has been suggested that HAD is more likely to occur in octogenarians and older adults. Women were less frequently affected by type ATAAD than men. On the other hand, women tend to be older than men at the time of diagnosis [26]. This is similar to the characteristics of HAD, and further research focusing on gender differences is needed. In addition, patients with HAD had a significantly longer duration of ventilator intubation, increased NPPV and CRRT usage, and a longer ICU stay. NPPV is used in cases of respiratory failure, such as pulmonary congestion and hypoxia [27]. In cases of unstable postoperative respiratory and circulatory dynamics, patients are forced to rest in bed, and they require long-term ICU management. Under these circumstances, postoperative delirium is more likely to develop and has been shown to be a risk factor for ADL reduction [28,29]. In elderly individuals, environmental changes due to hospitalisation may lead to decreased ADL independence and activity, resulting in HAD [6].

Early rehabilitation after surgery has recently been reported to be effective in preventing deterioration of physical function and ADL and in improving QOL [30,31]. In this study, progress in postoperative cardiac rehabilitation was significantly delayed in the patients with HAD. Immobility can cause physical deconditioning and muscle weakness. ADL has a strong correlation with muscle strength and physical performance [32], which could be explained by the fact that the mobility-related subitem scores of the BI tend to decrease. In addition, an early walking onset is likely to increase opportunities for ADL on the ward, such as toilet use and bathing, and to improve activity levels during hospitalisation. Therefore, we believe that early ambulation onset can have a preventive effect on physical function and ADL decline at the time of discharge. Critically ill postoperative patients with systemic inflammation and multiple organ dysfunction have been shown to experience a loss in muscle mass and physical functional decline [33]. It is considered that the delayed walking onset observed in this study was explained by hypercatabolism and immobility and tended to cause a severe decline in ADL. The perioperative period is important to manage sedation, agitation, and respiratory complications that tend to impede the progression of postoperative cardiac rehabilitation [34,35].

### 4.3. Study Limitations

First, this study excluded early transfer patients who had severe complications after surgery and who required preferential treatment in other departments. Therefore, critically ill patients with unstable general conditions, such as multiple organ failure, were not included, and the HAD-related factors in such patients are not clear. Moreover, some patients are transferred to the referral center after their condition has stabilized after the surgery. Second, this study adopted the definition of HAD. The BI evaluates various basic ADL, such as self-care and mobility [13]. Basic ADL does not only involve physical function such as muscle strength but also a wide variety of functional elements such as cognitive function and dexterity. The data on all these functional evaluations were not obtained for this study, and the functional elements that caused the BI decline were unclear.

In the future, authors should discuss these results and how they can be interpreted from the perspective of previous studies and the working hypotheses. The findings and their implications should be discussed in the possible broadest context. Future research directions may also be highlighted.

## 5. Conclusions

This study identified age, NPPV, postoperative delirium, and the timing of walking onset as relevant factors for HAD in patients who have undergone surgery for ATAAD. Furthermore, early walking onset was associated with a lower risk of HAD. Early rehabilitation based on appropriate criteria may prevent HAD, and perioperative management that enables early rehabilitation is important.

## Figures and Tables

**Figure 1 ijerph-19-12918-f001:**
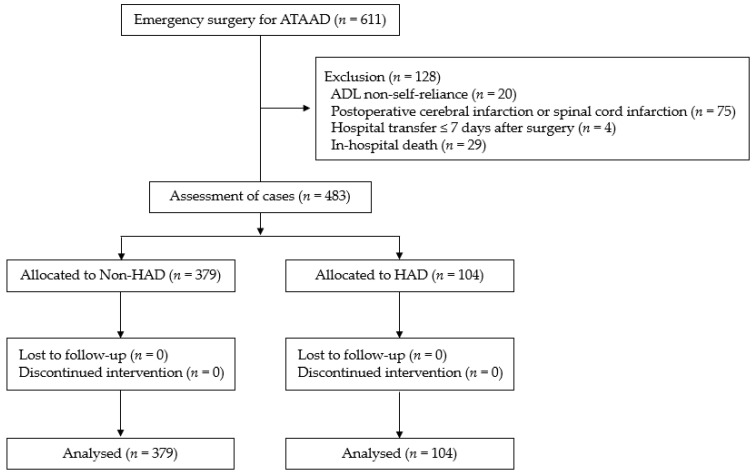
Schematic presentation of the study procedure. This study included 483 patients who required emergency surgery for ATAAD. ATAAD, acute type A aortic dissection; ADL, activities of daily living; HAD, mobilization on-associated disability.

**Figure 2 ijerph-19-12918-f002:**
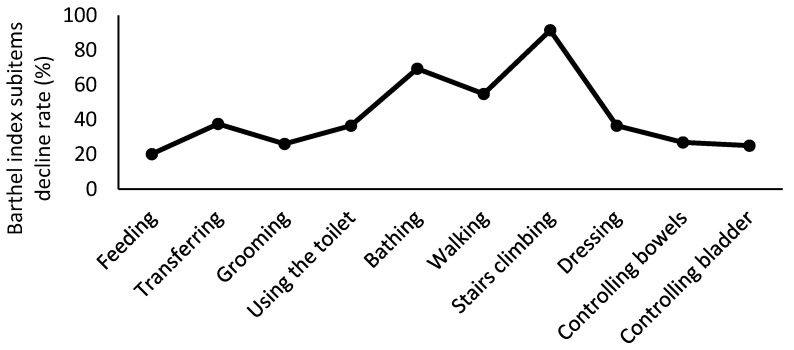
Subitems of reduced BI scores in patients with HAD (*n* = 104). Patients with declines in multiple subitems were included. BI: Barthel Index.

**Figure 3 ijerph-19-12918-f003:**
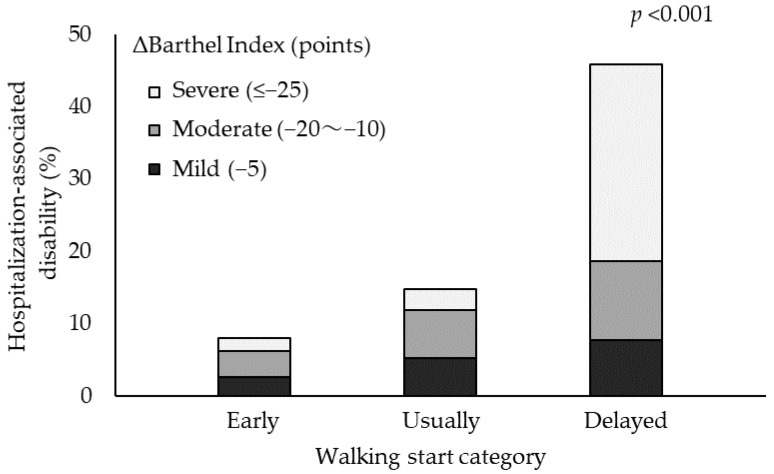
Barthel Index change by the walking start category. The decrease of −5 points was 33% in the early-start walking group, 44% in the usual-start walking group, and 20% in the delayed-start walking group; the decrease of −20 to −10 points was 44%, 33%, and 19%, respectively; and the decrease of −25 points or more was 22%, 22%, and 61%, respectively.

**Table 1 ijerph-19-12918-t001:** Clinical characteristics.

	Overall	Non-HAD	HAD	*p* Value
	(*n* = 483)	(*n* = 379)	(*n* = 104)	
Age, years	69 (59–78)	68 (58–75)	78 (66–83)	<0.001
Female, *n* (%)	250 (51.8)	187 (49.3)	63 (60.6)	<0.05
BMI, kg/m^2^	23.6 (21.3–26.3)	23.7 (21.4–26.4)	23.1 (21.2–26.0)	0.62
Comorbidity, *n* (%)				
Hypertension	336 (69.6)	260 (68.6)	76 (73.1)	0.38
Dyslipidaemia	93 (19.3)	71 (18.7)	22 (21.2)	0.58
Diabetes	25 (5.2)	17 (4.5)	8 (7.7)	0.19
Chronic kidney disease	24 (5.0)	17 (4.5)	7 (6.7)	0.35
Coronary artery disease	21 (4.3)	16 (4.2)	5 (4.8)	0.49
Chronic obstructive pulmonary disease	8 (1.7)	6 (1.6)	2 (1.9)	0.54
Blood biochemistry				
Haemoglobin, g/dL	12.3 (11.1–13.8)	12.5 (11.2–14.0)	11.6 (10.2–12.7)	<0.001
Platelet, 10^3^/μL	164 (132–200)	165 (132–202)	161 (133–194)	0.53
Creatinine, mg/dL	0.87 (0.70–1.12)	0.86 (0.69–1.12)	0.91 (0.75–1.13)	0.15
Albumin, g/dL	3.6 (3.3–4.0)	3.7 (3.4–4.0)	3.4 (3.0–3.7)	<0.001
CRP, mg/dL	0.15 (0.04–0.99)	0.13 (0.04–0.90)	0.18 (0.05–1.40)	0.38
Aortic surgery, *n* (%)				0.40
Ascending aortic replacement	255 (51.6)	187 (49.2)	68 (59.6)	
Hemi arch replacement	26 (5.3)	18 (4.7)	8 (7.0)	
Total arch replacement	135 (27.3)	112 (29.5)	23 (20.2)	
Concomitant	78 (15.8)	63 (16.6)	15 (13.2)	
Operation time, min	225 (185–287)	225 (182–285)	229 (193–304)	0.21
Cardiopulmonary bypass time, min	128 (105–168)	128 (104–168)	128 (110–169)	0.54
Circulatory arrest time, min	27 (22–38)	27 (21–39)	28 (23–38)	0.42
IMV time, h	22.9 (18.2–34.6)	22.1 (17.7–30.9)	31.6 (21.8–46.3)	<0.001
Bleeding, ml	220 (150–350)	210 (150–320)	250 (163–438)	<0.05
APACHE II score, points	15 (12–18)	15 (12–18)	16 (13–18)	<0.05
NPPV, *n* (%)	102 (21.1)	67 (17.7)	35 (33.7)	<0.001
CRRT, *n* (%)	12 (2.5)	6 (1.6)	6 (5.8)	<0.05
Acute kidney injury, *n* (%)	175 (36.2)	130 (34.3)	45 (43.3)	0.09
Delirium, *n* (%)	123 (25.5)	71 (18.7)	52 (50.0)	<0.001
Distal enlargement, *n* (%)	34 (7.0)	26 (6.7)	8 (7.7)	0.77
ICU stay, days	3 (2–5)	3 (2–4)	4 (3–7)	<0.001
Hospital stay, days	14 (11–21)	14 (11–21)	13 (10–22)	0.50
Cardiac rehabilitation, days				
Sitting	2 (1–2)	2 (1–2)	2 (2–3)	<0.001
Standing	2 (1–3)	2 (1–2)	2 (2–3)	<0.001
Walking	3 (3–4)	3 (2–4)	5 (3–7)	<0.001
Barthel index, points				
Preoperative	100 (100–100)	100 (100–100)	100 (100–100)	0.40
Discharge	100 (100–100)	100 (100–100)	80 (55–90)	<0.001
Home discharge, *n* (%)	276 (57.1)	255 (67.3)	21 (20.2)	<0.001

BMI, body mass index; CRP, C-reactive protein; IMV, invasive mechanical ventilation; APACHE, Acute Physiology and Chronic Health Evaluation; NPPV, noninvasive positive pressure ventilation; CRRT, continuous renal replacement therapy; ICU, intensive care unit.

**Table 2 ijerph-19-12918-t002:** Noncompletion of early mobilisation.

	*n* = 131
Not awakening	51 (38.9)
Respiratory-related	26 (19.8)
Sedation	14 (10.7)
Uncontrolled blood pressure	10 (7.6)
Arrhythmia	8 (6.1)
Acute kidney injury	6 (4.6)
Pain distress	5 (3.8)
Active bleeding	4 (3.1)
Acute limb ischemia	2 (1.5)
Others	5 (3.8)

**Table 3 ijerph-19-12918-t003:** Clinical characteristics according to walking onset category.

	Walking Onset Category		
	Early	Usual	Delayed
	(*n* = 111)	(*n* = 243)	(*n* = 129)
Age, years	66 (55–75)	71 (62–79) ^1^	69 (57–81)
Female, *n* (%)	48 (43.2)	137 (56.4)	65 (50.4)
BMI, kg/m^2^	23.4 (21.3–25.6)	23.7 (21.5–26.4)	23.1 (20.9–26.6)
Comorbidity, *n* (%)			
Hypertension	79 (71.2)	170 (70.0)	87 (67.4)
Dyslipidemia	22 (19.8)	50 (20.6)	21 (16.3)
Diabetes	4 (3.6)	15 (6.2)	6 (4.7)
Chronic kidney disease	3 (2.7)	11 (4.5)	10 (7.8)
Coronary artery disease	2 (1.8)	12 (4.9)	7 (5.4)
Chronic obstructive pulmonary disease	3 (2.7)	3 (1.2)	2 (1.6)
Operation time, min	215 (184–280)	218 (180–268)	263 (197–324) ^1,2^
Cardiopulmonary bypass time, min	123 (102–165)	125 (103–157)	149 (110–192) ^1,2^
IMV time, h	19.3 (16.4–23.7)	22.6 (18.0–30.5) ^1^	38.1 (22.2–50.7) ^1,2^
Bleeding, ml	200 (130–270)	220 (150–350)	280 (165–453) ^1,2^
APACHE II score, points	14 (12–17)	15 (13–17)	15 (13–19) ^1^
NPPV, *n* (%)	8 (7.2)	42 (17.3)	52 (40.3) ^3^
CRRT, *n* (%)	1 (0.9)	2 (0.8)	9 (7.0) ^3^
Delirium, *n* (%)	11 (9.9)	58 (23.9)	54 (41.9) ^3^
ICU stay, days	2 (1–2)	3 (2–3) ^1^	5 (4–8) ^1,2^
Hospital stay, days	13 (10–18)	14 (11–20)	19 (13–28) ^1,2^
Preoperative BI, points	100 (100–100)	100 (100–100)	100 (100–100)

^1^*p* < 0.05 vs. Early, ^2^
*p* < 0.05 vs. Usually, ^3^
*p* < 0.05 vs. χ^2^ test. BMI, body mass index; IMV, invasive mechanical ventilation; APACHE, Acute Physiology and Chronic Health Evaluation; NPPV, noninvasive positive pressure ventilation; CRRT, continuous renal replacement therapy; ICU, intensive care unit; BI, Barthel index.

**Table 4 ijerph-19-12918-t004:** Univariate and multivariate analyses for hospitalisation-associated disability.

	Univariate			Multivariate		
	OR	95% CI	*p* Value	OR	95% CI	*p* Value
Age [every 1-year increase]	1.05	1.03–1.08	<0.001	1.05	1.02–1.09	<0.01
<60 years	1.00			1.00		
60–69 years	0.75	0.36–1.57	0.45			
70–79 years	1.38	0.71–2.67	0.35			
≥80 years	4.83	2.56–9.12	<0.001	4.28	1.62–11.35	<0.01
Female	1.58	1.01–2.45	<0.05			
Albumin [every 1 g/dL increase]	0.37	0.23–0.58	<0.001			
Haemoglobin [every 1 g/dL increase]	0.76	0.67–0.86	<0.001			
APACHE II score [every 1-point increase]	1.05	1.00–1.11	0.07			
IMV time [every 1-h increase]	1.03	1.02–1.05	<0.001			
NPPV	2.36	1.46–3.84	<0.01	2.15	1.10–4.19	<0.05
CRRT	3.81	1.20–12.1	<0.05			
Delirium	4.34	2.73–6.89	<0.001	2.93	1.60–−5.37	<0.01
ICU stay (every 1-day increase)	1.28	1.18–1.39	<0.001			
Walking (every 1-day increase)	1.37	1.23–1.54	<0.001	1.29	1.07–1.56	<0.01

APACHE, Acute Physiology and Chronic Health Evaluation; IMV, invasive mechanical ventilation; NPPV, noninvasive positive pressure ventilation; CRRT, continuous renal replacement therapy; ICU, intensive care unit.

**Table 5 ijerph-19-12918-t005:** Univariate and multivariate analyses for hospitalisation-associated disability by walking onset category.

	Univariate			Multivariate *		
	OR	95% CI	*p* Value	OR	95% CI	*p* Value
Walking start category						
Early	1.00			1.00		
Usual	1.97	0.91–4.25	0.08	0.97	0.42–2.26	0.94
Delayed	9.55	4.45–20.52	<0.001	2.76	1.05–7.21	<0.05

* Adjusted for age, sex, albumin, haemoglobin, duration of ventilator intubation, APACHE II score, CRRT, NPPV, delirium, and length of stay in the ICU.

## Data Availability

The datasets used and analysed during the current study are available from the corresponding author upon reasonable request.

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
