# Peer review of "Factors Related to Hospitalisation-Associated Disability in Patients after Surgery for Acute Type A Aortic Dissection: A Retrospective Study"

_ijerph, 2022, doi:10.3390/ijerph191912918_

Round 1

Reviewer 1 Report

This is well-written original research about HAD after surgery for type A acute aortic dissection. 

For completeness, you should add the definition of acute aortic dissection and if you consider also the traumatic type A acute aortic dissection.

Furthermore, HAD is more present in females, but it is not emphasized in the discussion, although there is a lot of evidence for gender diversity in this pathology (es Bossone E, Carbone A, Eagle KA. Gender Differences in Acute Aortic Dissection. J Pers Med. 2022 Jul 15;12(7):1148. doi: 10.3390/jpm12071148).

Reviewer 2 Report

Lines 27-28 - They are repetitive

Table 3 - p-values are missing for certain characteristics 

Lines 133-134 - You can clarify the statement to better reflect that BI scores saw a greater decline in categories of stair climbing, bathing and walking 

Line 167 - "than that had by the early-start walking group" - can be changed to "than early-start walking group" for easier readability 

Lines 182-183 - related to reference 1. It is controversial to say patients with HAD (hospital-associated disability) need strict blood pressure control. It should be the case with all patients irrespective of HAD or non-HAD

Lines 195-197 - Please revise the statement from a language standpoint. You have described factors of ATAAD as well as surgical complications in one sentence which can be confusing to readers. 

Line 221 - Incidence of HAD has been reported as low. Low when compared to what? If comparing to acute medical illness, I do not think it is an appropriate comparison. 

Line 238 - replace 'influenced' with 'explained' 

Reviewer 3 Report

Dear Authors,

Please find attached your manuscript with my comments and edits. I provided suggestions to organize the content of the manuscript in order to allow a reader to follow the text much easier

Best regards,
